# The Effect of Probiotic Yogurt Containing Lactobacillus Acidophilus LA-5 and Bifidobacterium Lactis BB-12 on Selected Anthropometric Parameters in Obese Individuals on an Energy-Restricted Diet: A Randomized, Controlled Trial

**Katarzyna Banach**[ID]**, Paweł Glibowski ***[ID] **and Paulina Jedut**

Department of Biotechnology, Microbiology and Human Nutrition, University of Life Sciences in Lublin, Skromna 8, 20-704 Lublin, Poland; katarzynabanach_22@wp.pl (K.B.); paulinajedut1@wp.pl (P.J.)

\* Correspondence: pawel.glibowski@up.lublin.pl

**Abstract:** Previous studies using probiotics have shown strain-dependent effects on body mass index (BMI), body mass, or fat mass (FM). The aim of this study was to evaluate how the addition of yogurt containing *Lactobacillus acidophilus* LA-5 and *Bifidobacterium animalis* subsp. *lactis* BB-12 strains to a diet plan affects selected anthropometric parameters in obese people on an energy-restricted diet. Fifty-four subjects aged 20–49 (34.52 ± 9.58) years were included in this study. The recruited subjects were assigned to two subgroups: consuming probiotic yogurt along with a hypocaloric diet (GP) (n–27) or the same diet but without an intentional introduction of yogurt (GRD) (n–27) for 12 weeks. Both GP and GRD decreased body weight, BMI, fat mass and visceral fat by 5.59 kg and 4.71 kg, 1.89 and 1.61 kg/m$^2$, 4.80 kg and 4.07 kg, and 0.68 and 0.65 L, respectively, although the obtained differences were not significant. Analysis of GP and GRD results separately at the beginning and end of the intervention showed that fat loss was substantial in both groups ($p < 0.05$). Consumption of yogurt containing LA-5 and BB-12 does not significantly improve anthropometric parameters in obese patients.

**Keywords:** probiotics; obesity; probiotic yogurt; energy-restricted diet; body composition

## 1. Introduction

The growing phenomenon of overweight and obesity has led to a global problem. Current data estimate that 650 million people are obese worldwide, while 1.9 billion are overweight [1]. The latest trend analyses show that the number of obese or overweight people is steadily increasing both in Europe and around the world [2,3]. The obesity disease is characterized by excessive fat accumulation, which is a consequence of a long-term energy surplus. The reasons given for this state are primarily seen in excessive energy consumption, reduced energy expenditure, or both at the same time. Additionally, changes in eating habits and the increased availability of high-calorie foods have made overweight and obesity one of the most severe health problems of our era. Obesity is a multifactorial disease caused by both environmental and genetic determinants. Importantly, overweight and obesity positively correlate with other chronic diseases such as type 2 diabetes, cardiovascular diseases, some types of cancer, and infertility. Annually, nearly 2.8 million deaths in the world are a consequence of excess weight [1,4,5].

Scientific reports from the last several years have investigated the relationship between the composition of the intestinal microbiota and the development of obesity [6]. People with excessive

body weight have a higher positive proportion of *Firmicutes* to *Bacteroidetes* than normal-weight people. It seems that significant variation within *Firmicutes* may contribute to more efficient energy extraction from food sources and increase energy storage in host fat tissue [7,8]. Intestinal bacteria ferment indigestible carbohydrate residues, synthesize short-chain fatty acids and amino acids, and thus can contribute to an increase in the amount of energy supplied to the host. However, it is not determined whether this process is of clinical importance [9,10]. On the other hand, by-products of bacterial fermentation can reduce appetite and increase satiety [11] as well as decrease body weight and Body Mass Index (BMI) [12]. Consequently, the intestinal microbiota is a potentially modifiable factor associated with the prevention and treatment of excessive body weight.

Probiotic bacteria are described as 'live microorganisms which when administered in adequate amounts confer a health benefit on the host' [13]. Probiotic food supplementation has grown in popularity in recent years [14]. The best known probiotic microorganisms include strains belonging to the genera *Lactobacillus* and *Bifidobacterium* [15]. *Bifidobacterium animalis* subsp. *lactis* BB-12 has been used in clinical trials alone and with other bacteria such as *Lactobacillus acidophilus* LA-5 or *Streptococcus thermophilus* [16]. The results of human experiments provide the basis for attributing several beneficial effects to LA-5 and BB-12. They show, among others, prophylactic activity against children rotavirus diarrhea infection in children [17], and also reduce the incidence of skin eczema [18]. Many studies were carried out with the use of probiotic preparations, although probiotic food products containing LA-5 and BB-12 strains were also studied. Consumption of the above-mentioned products reduced inflammation markers [19] and improved blood lipid profile [20]. In this context, probiotics can be a useful tool in the treatment of obesity and its complications.

Regular intake of drugs or dietary supplements containing probiotics can be problematic due to their often high price, the pharmacotherapy used by the patient, or frequent irregularities on the dietary supplement market [21]. For this reason, it is worth focusing on widely available food enriched with probiotic strains. The probiotic product should contain at least $10^6$–$10^7$ of viable probiotic bacteria cells per one ml of product [22].

To the best of our knowledge, there is no research describing whether the dietary addition of low-fat yogurt containing LA-5 and BB-12 will show a positive effect on body composition in overweight and obese women and men on a balanced hypocaloric diet.

According to the information given above, the purpose of the study was to assess the effect of widely available yogurts containing *Lactobacillus acidophilus* LA-5 and *Bifidobacterium lactis* BB-12 strains on selected anthropometric parameters in obese subjects on an energy-restricted diet.

## 2. Materials and Methods

### 2.1. The Test Group

The study was carried out from February 2017 until May 2018 on 54 patients of the Dietitian Service, a unit of the University of Life Sciences in Lublin. The subjects were aged 20–49. Among the study group, 35% (n–19) were men, and 65% (n–35) were women. The basis of the allocation was that one member of the control group was matched to a member of the experimental group given the three main independent variables important for the study: sex, age, and BMI value. At first, pairs of participants were selected based on convergent scores on a particular measure of independent variables, and then each member of the pair was randomly assigned to two subgroups: consuming probiotic yogurt along with a hypocaloric diet (GP) or the same diet but without an intentional introduction of yogurt (GRD). Sealed envelopes were used to implement the random allocation of participants. This process was performed by one person (K.B.). This study followed the guidelines set out in the Consolidated Standards of Reporting Trials (CONSORT) [23]. Statistical analysis confirmed the lack of significant differences in the assessment of the height of subjects in particular subgroups. The inclusion criteria for the study included: age 18–50, obesity grade 0 according to the American Association of Clinical Endocrinologists (AACE) and American College of Endocrinology (ACE) classification [24],

and readiness to comply with energy-restricted diet principles. The exclusion criteria were: occurrence of obesity complications listed in the AACE/ACE classification, other metabolic and autoimmune disorders, infectious diseases during the study period, pregnancy, lactation, taking medications that may affect body weight, lactose intolerance, and allergy to cow's milk. Two participants became pregnant during the study and were therefore excluded from the trial. Individual cases involving health issues also forced some drop-outs from the study (Figure 1). All subjects gave their informed consent for inclusion before they participated in the study. The study was conducted in accordance with the Declaration of Helsinki, and the protocol was approved by the Ethics Committee of Medical University of Lublin. Decision number: KE-0254/180/2019.

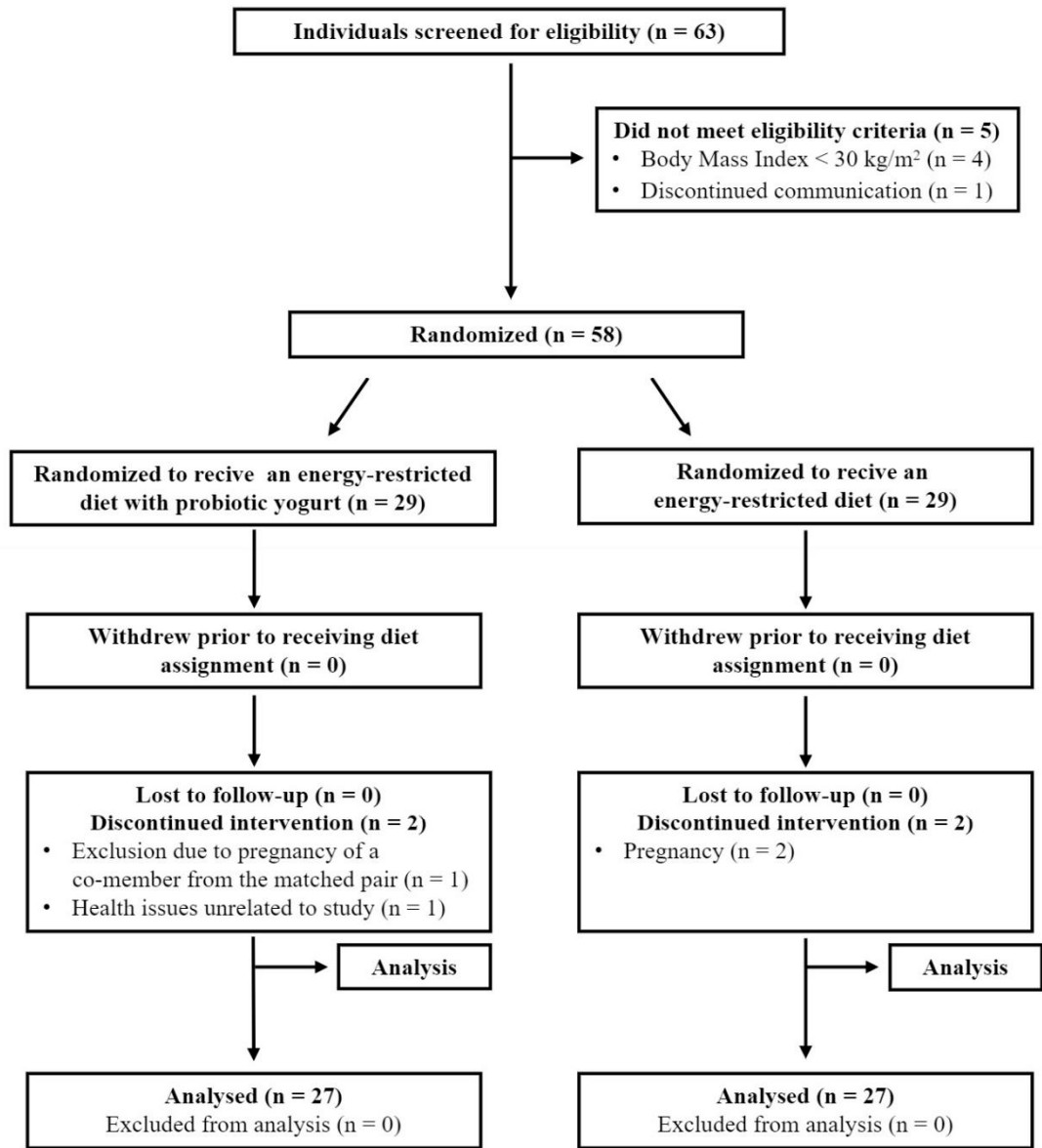

**Figure 1.** Flow diagram of patient recruitment and randomization process. Based on the Consolidated Standards of Reporting Trials (CONSORT) 2010 flow diagram [23].

## 2.2. Anthropometric Measurements

Anthropometric measurements of all participants were made at the starting point and after 12 weeks of the study duration. Body height in a standing position was measured through the SECA 216 wall mounted stadiometer with an accuracy of 0.1 cm. The waist circumference was

determined halfway between the lower edge of the rib arch and the upper iliac crest, using SECA 201 metric tape. Body mass was recorded in lightweight clothing on a digital scale with 0.1 kg accuracy. To assess the relative body weight, BMI with the adult classification recommended by the World Health Organization (WHO) was used. The BMI value was calculated based on the quotient of the measured body weight (kg) and body height (m) squared. An SECA mBCA515 analyser was applied for the body composition analysis using the bioelectric impedance method. The measurement was carried out using the eight-point method.

### 2.3. Research Assumptions and Intervention Design

Previous trials assessing the effects of probiotics or dairy products on weight loss indicated that consuming probiotics could reduce body weight and BMI, with a potentially greater effect when the duration of intervention was longer than 8 weeks, or the objects were had excessive body weight [12]. In this aspect, the selection of the appropriate probiotic strain may raise doubts, because nowadays we cannot discern the relevance of specific subspecies or strains in the control of adiposity. For these reasons, we decided to analyze the strains LA-5 and BB-12. The research hypothesis was: consumption of yogurt containing LA-5 and BB-12 significantly improves anthropometric parameters in obese patients on an energy-restricted diet.

In the construction of the above hypothesis, we assumed that a study group is homogenous in the context of dietary compliance and a significance level of $p < 0.05$ indicating statistically significant differences.

This randomized, single-blind study is designed to evaluate the introduction of low-fat yogurt to a diet plan containing *Lactobacillus acidophilus* LA-5 and *Bifidobacterium lactis* BB-12 strains on selected anthropometric parameters in obese people on an energy-restricted diet. Fifty-four subjects eligible for the study were randomly assigned to one of two groups: those consuming probiotic yogurt (250 g/d) along with a hypocaloric diet (GP) (n–27) or those on the same hypocaloric diet without deliberate addition of yogurt (GRD) (n–27) for 12 weeks. The probiotic yogurt (Tola + wapń, OSM Krasnystaw, Krasnystaw, Poland) was a commercially available product prepared with the using the starter cultures of *Streptococcus salivarius* subsp. *thermophilus* and *Lactobacillus delbrueckii* subsp. *bulgaricus*, enriched with the probiotic culture of two strains: *Lactobacillus acidophilus* LA-5 and *Bifidobacterium lactis* BB-12. One hundred grams of probiotic yogurt contained 244 kJ (58 kcal), 4.3 g protein, 5.7 g carbohydrates, and 2.0 g fat. Additionally, yogurt was enriched with calcium in an amount of 240 mg/100 g of product (manufacturer's data).

### 2.4. Dietary Intervention

In this study, the dietary intervention was designed on the bases of guidelines of the Polish Dietetic Association [25]. The energy deficit used was in the range of 500–800 kcal/d, which would be expected to lead to weight loss rates of around 0.5–0.8 kg/week. The study lasted 12 weeks. The distribution of macronutrients was 15–25% of the energy supplied from proteins, 25–35% from fats, and 45–60% from carbohydrates. An individual nutrition program for each participant was prepared based on information obtained from the nutrition interview conducted at the first visit and data from five-day food diaries prepared by the patient before the first visit. There were no significant differences in the supply of macronutrients, vitamins, and minerals in both groups. The only difference was the daily consumption of probiotic yogurt by GP. The prepared nutritional plans did not include other probiotic products during the intervention. During the day, participants consumed 4 or 5 meals, depending on their preferences. Moreover, the plans prepared for each participant in the study were based on products with the least degree of processing, minimizing the supply of simple carbohydrates, saturated fatty acids, and trans fatty acids. Attention was paid to high consumption of vegetables and fruits, high quality products, and meals prepared at home. Participants were also encouraged to increase physical activity, which, combined with the energy deficit, promotes a more favorable body recomposition than a reduction diet alone [26]. The minimum goal was to reach 150 min of moderate activity or

75 min of high-intensity exercise during the week [27]. The intervention also included behavioral aspects. Participants were encouraged, for example, to set goals, to strengthen their motivation, or take advantage of environmental support (household members) [23].

At least once a month, participants were required to attend a follow-up visit, during which the course of the experiment was monitored, difficulties arising from compliance with the diet discussed, adjustments made to diet plans, and anthropometric measurements performed. The respondents were asked in detail about the diet, eating yogurt, and doing physical activity. The participants had unlimited access to telephone or online support from a qualified dietitian if needed.

*2.5. Statistical Analysis*

The obtained test results were subjected to statistical analysis. Values of the measurable parameters are presented as the mean value and standard deviation, while non-measurable parameters are presented with the number and percentage. For measurable features, the distribution normality of the analyzed parameters was evaluated using the Shapiro–Wilk test. Student's t-test was used to compare the two average values, while for the non-parametric values, the Mann–Whitney U test was used without assuming that the values are normally distributed in the studied parameter. A significance level of $p < 0.05$ was assumed, indicating statistically significant differences. All data were analyzed by the STATISTICA 13.0 computer software (StatSoft, TIBCO Software, Palo Alto, CA, USA, 2019). The diet plans were analyzed using the ALIANT diet calculator (Cambridge Diagnostics, Anmarsoft, Gdańsk, Poland, 2018).

## 3. Results

All participants completed the study, but four subjects were excluded from the statistical analysis due to circumstances such as pregnancy, health problems, and others (Figure 1). No adverse effects related to the consumption of probiotic yogurt and (or) the implementation of a reduction diet were reported during the experiment. Descriptive characteristics and comparison within the two groups of subjects included in the study are shown in Table 1. At the beginning of the experiment, there were no statistically significant differences in anthropometric characteristics between the observational groups and between the number of participants who completed the study.

**Table 1.** Characteristics of the studied group before the intervention.

| Parameter | GP ($n = 27$) Women ($n = 17$), men ($n = 10$) | GRD ($n = 27$) Women ($n = 18$), men ($n = 9$) | *p*-Value |
|---|---|---|---|
| Age (years) | 34.85 ± 9.22 | 34.18 ± 10.80 | 0.64662 |
| Height (m) | 1.71 ± 0.10 | 1.72 ± 0.09 | 0.63568 |
| Body Weight (kg) | 104.37 ± 20.93 | 101.25 ± 14.12 | 0.52382 |
| BMI (kg/m$^2$) | 35.55 ± 4.40 | 34.21 ± 3.15 | 0.20520 |
| Fat Mass (kg) | 43.72 ± 9.13 | 41.46 ± 8.35 | 0.44653 |
| Visceral Adipose Tissue (VAT) (L) | 4.01 ± 2.74 | 3.22 ± 1.98 | 0.41122 |
| Free Fat Mass (kg) | 60.65 ± 15.01 | 59.79 ± 10.33 | 0.76868 |
| Muscle Mass (kg) | 29.93 ± 8.37 | 29.43 ± 6.07 | 0.86265 |
| Total Body Water (L) | 45.11 ± 10.86 | 44.51 ± 7.38 | 0.84230 |
| Phase Angle (°) | 5.57 ± 0.59 | 5.61 ± 0.63 | 0.88309 |

*p*-value—statistically significant difference between GP and GRD groups at $p < 0.05$.

The qualitative analysis of nutritional plans used during interventions is presented in Table 2. Energy levels and consumption of the primary nutrients were similar in both groups. Other important components of a balanced diet such as polyunsaturated fatty acids or dietary fiber were also identical in GP and GRD. The energy value of the diet was approximately 1600 kcal/d for women and 2100 kcal/d for men, which gave an average value of 1850 kcal/d. The energy deficit applied, depending on the case, was 500–800 kcal/d. The presented results meet the assumptions of a balanced weight-loss diet.

**Table 2.** Daily consumption of energy and selected nutrients by the studied groups.

| Parameter | GP (*n* = 27) Women (*n* = 17), Men (*n* = 10) | GRD (*n* = 27) Women (*n* = 18), Men (*n* = 9) | *p*-Value |
|---|---|---|---|
| Energy (kcal/day) | 1843.46 ± 270.02 | 1824.69 ± 257.46 | 0.64662 |
| Carbohydrate (g/day) | 230.37 ± 35.94 | 228.90 ± 37.53 | 0.84907 |
| Carbohydrate (%) | 45.83 ± 2.59 | 45.79 ± 2.34 | 0.94312 |
| Protein (g/day) | 109.44 ± 14.06 | 110.20 ± 20.78 | 0.95860 |
| Protein (%) | 23.84 ± 1.69 | 24.08 ± 2.42 | 0.73585 |
| Total Fat (g/day) | 61.99 ± 10.67 | 60.77 ± 6.43 | 0.94483 |
| Total Fat (%) | 30.23 ± 2.32 | 30.22 ± 2.70 | 0.98281 |
| Dietary Fiber (g/day) | 37.85 ± 7.72 | 38.59 ± 5.61 | 0.68687 |
| Saturated Fatty Acids (g/day) | 14.45 ± 4.17 | 15.74 ± 5.4 | 0.45693 |
| n-3 Fatty Acids (g/day) | 2.47 ± 1.03 | 2.12 ± 0.68 | 0.34575 |
| n-6 Fatty Acids (g/day) | 9.96 ± 3.45 | 9.42 ± 3.74 | 0.56221 |

*p*-value—statistically significant difference between GP and GRD group at $p < 0.05$.

Table 3 presents the results of anthropometric measurements for two groups at the beginning and end of the 12-week intervention. The differences were compared within each group separately (at week 0 and 12 of the study) and between GP and GRD at the endpoint of the experiment.

**Table 3.** Anthropometric characteristics in GP and GRD groups before and after 12 weeks of intervention.

| Parameter | GP (*n* = 27) | | | GRD (*n* = 27) | | | *p*-Value |
|---|---|---|---|---|---|---|---|
| | Week 0 | Week 12 | Difference | Week 0 | Week 12 | Difference | |
| Body weight (kg) | 104.37 ± 20.93 | 98.78 ± 20.01 | −5.59 | 101.25 ± 14.12 | 96.54 ± 13.25 | −4.71 | 0.62995 |
| BMI (kg/m2) | 35.55 ± 4.40 | 33.65 ± 4.28 | −1.89 | 34.21 ± 3.15 | 32.60 ± 3.04 | −1.61 | 0.30169 |
| Fat mass (kg) | 43.72 ± 9.13 | 38.92 ± 8.89 | −4.80* | 41.46 ± 8.35 | 37.39 ± 7.66 | −4.07 * | 0.50138 |
| VAT (L) | 4.01 ± 2.74 | 3.33 ± 2.44 | −0.68 | 3.22 ± 1.98 | 2.57 ± 1.58 | −0.65 | 0.47813 |
| Free fat mass (kg) | 60.65 ± 15.01 | 59.82 ± 14.48 | −0.83 | 59.79 ± 10.33 | 59.15 ± 10.71 | −0.64 | 0.66537 |
| Muscle mass (kg) | 29.93 ± 8.37 | 28.89 ± 8.67 | −1.04 | 29.43 ± 6.07 | 28.88 ± 6.13 | −0.55 | 0.72934 |
| Total body water (L) | 45.11 ± 10.86 | 44.12 ± 10.75 | −0.99 | 44.51 ± 7.38 | 43.87 ± 7.66 | −0.64 | 0.65285 |
| Phase angle (°) | 5.57 ± 0.59 | 5.61 ± 0.65 | 0.04 | 5.61 ± 0.63 | 5.61 ± 0.60 | 0.00 | 0.96553 |

*p*-value—statistically significant difference between GP and GRD group at $p < 0.05$. * statistically significant difference between week 0 and week 12 in each group at $p < 0.05$.

As expected, weight loss was observed in both groups, although no significant ($p < 0.05$) differences were found between the groups. However, it is worth emphasizing that for both groups a loss of initial body weight of 3–5% was achieved, which reduces the risk of lifestyle diseases [28]. The majority of the total body weight loss was fat, and this reduction was significant for both groups. The addition of probiotic yogurt to the diet plan did not contribute to the occurrence of statistically significant differences between GP and GRD in fat reduction. Similarly to the decrease in total body weight, the BMI index in both groups decreased. Excessive accumulation of visceral fat (VAT) leads to visceral obesity. According to Peine et al. [29], normal VAT volumes are up to 1.2 L for women and 2.1 L for men. Values between 1.2 and 1.9 L for women and between 2.1 and 3.8 for men indicate an increased but still acceptable VAT content, while higher volumes mean that a VAT level is too high. In both groups, the overall volume of VAT decreased at a similar level, and even after 12 weeks of intervention, most participants had an increased or high volume of visceral fat. There was also a slight loss of free fat mass (including muscle mass and total water), which is typical during weight reduction [30]. The biological significance of the phase angle is not fully known, but it is considered an indicator of cell health. A higher phase angle value correlates with better cell function. Studies conducted to date to determine the range of population norms of phase angle values indicate that the phase angle value in healthy adults is 5–7°, and a value below 5.0° indicates malnutrition [31,32]. The obtained results indicate a satisfactory state of the cells in the studied group. Using an energy-restricted diet and (or), the addition of probiotics to the diet plan did not change this indicator.

## 4. Discussion

The growing interest in research on the role of intestinal microbiota in obesity and the view that modification in the composition of intestinal bacteria can help achieve permanent weight loss [33] prompted us to conduct this experiment. The main aim of this study was to investigate the effect of consuming yogurt containing *Lactobacillus acidophilus* LA-5 and *Bifidobacterium lactis* BB-12 strains on selected anthropometric parameters in obese people on an energy-restricted diet. To the best of our knowledge, an intervention consisting of inserting yogurt with the strains mentioned above into the hypocaloric diet with a comparison of the results in regard to the effects of a balanced reduction diet without yogurt has not previously been conducted. The practical aspect of the study is manifested in the fact that the used yogurt is a product widely available, often used in nutritional plans devised by dieticians.

The basic approach in anti-obesity diet therapy is to lead the patient to an energy deficit, which should cause weight loss. The practical goal is to obtain and maintain long-term body weight reduced by 5–10% of the current weight. In patients with a high risk of cardiovascular disease, a reduction and maintenance of body weight by 3–5% brings a significant improvement in lipid and carbohydrate metabolism [28,34]. As Foster et al. [35] showed, this effect would not be satisfactory for obese patients, because their common perceptions about the desired weight loss oscillate around -38.4%, which is often not possible to achieve. That is why it is worth paying attention to the fact that even a small reduction in body weight brings positive health effects. Unfortunately, only 20% of people who have reduced their body weight manage to maintain the achieved effect for at least another year [36]. For this reason, it is worth looking for solutions that will help in long-term weight management. One such strategy that has been of interest to scientists for years may be the introduction of probiotics into the diet.

Some reports suggest that the inclusion in the diet of probiotic strains of the *Lactobacillus* genus may have beneficial effects on reducing weight and improving body composition both through the use of probiotic preparations and food products after 12 weeks of application [37,38]. It has been shown that the inclusion of *Lactobacillus gasseri* SBT2055 or BNR17 and *Lactobacillus amylovorus* CP1563 in overweight adults seems to be justified, as they may promote a reduction in overall body weight. Similar effects were noted for *Lactobacillus plantarum* TENSIA and *Lactobacillus rhamnosus* CGMCC13724 in combination with a hypocaloric diet. Furthermore, combinations of *Lactobacillus plantarum* KY1032 with *Lactobacillus curvatus* HY7601, and *Lactobacillus acidophilus* LA-14 with *Lactobacillus casei* LC-11 and phenolic compounds can contribute to obtaining similar results [39]. On the other hand, Agerholm-Larsen et al. [40] noted weight gain using *Lactobacillus rhamnosus* StLr and *Lactobacillus acidophilus* StLa. However, this could be due to the inclusion of yogurt in the participant's diet leading to an energy surplus. Furthermore, in the study using *Lactobacillus reuteri* JBD30l in intervention, weight gain was noted among the participants; however, the probiotic group gained less weight than the control [41]. One possible explanation for this is that *Lactobacillus reuteri* can improve the ability to absorb and process nutrients in the intestine [42].

Several teams of researchers [19,43–45] have already undertaken interventions with LA-5 and BB-12, but their research designs were different from ours, and anthropometric parameters were a secondary assessment. This study showed no significant difference in the progression of weight reduction between GP and GRD groups. The results obtained are in line with the conclusions drawn by Madjd et al. [43], who used yogurt with LA-5 and BB-12 and a placebo on a group of obese women. In the cited experiment, it was not observed that the consumption of probiotics contributes to the occurrence of significant differences ($p$-value = 0.248) in weight reduction between two groups on a reducing diet. However, the authors suggest that the intervention used may have a positive effect on the lipid profile and insulin sensitivity in obese women during the weight-loss stage. Zarrati et al. [19] verified the hypothesis that probiotics (LA-5, *Lactobacillus casei* DN001, and BB-12) might have beneficial effects in alleviating inflammation and obesity complications, and acting synergistically with a healthy reducing diet. Moreover, although the authors showed that consuming probiotic yogurt with or without a reduction diet for 8 weeks had a positive effect on the immune system in overweight and

obese subjects, it was not noticed that the used probiotic strains significantly improved anthropometric parameters such as body weight, BMI value or circumference waist in the studied group. Furthermore, in the Savard et al. [44] experiment, one of the assessed effects of yogurt supply from LA-5 and BB-12 for 4 weeks was changed in body weight and waist circumference in people with normal BMI. As in the case of the reports mentioned above, this experiment did not show the significant role of probiotic bacteria in changing anthropometric parameters in the studied group. The observed differences in body weight change ranged 0.1–0.5% (*p*-value = 0.45) depending on the group. Xavier-Santos et al. [45] evaluated the effect of drinking a synbiotic mousse containing LA-5 and prebiotics (inulin and fructooligosaccharides) on blood pressure as well as anthropometric and biochemical measurements of obese volunteers with metabolic syndrome. All evaluated parameters were measured at the beginning and after 8 weeks of intervention. In the cited study, none of the parameters assessed changed significantly in both the synbiotic and placebo groups. This was convergent with the results of our experiment and does not give grounds to claim that the probiotic yogurt with LA-5 and BB-12 strains significantly contribute to the reduction in parameters such as body weight, BMI, or VAT in obese people.

The above-mentioned state of affairs may result from the fact that inconvenient conditions of the gastrointestinal tract (e.g., low pH, digestive agents present in the gastric and pancreatic secretions, presence of bile, etc.), may have affected the LA-5 functionality and survival. Some reports demonstrated the low tolerance of LA-5 to artificial gastrointestinal juice in an assay that simulated the gastrointestinal tract conditions [45,46], which could explain why yogurt containing LA-5 and BB-12 failed to improve anthropometric parameters in obese patients in our study. On the other hand, BB-12 clearly shows high gastric acid and bile tolerance compared to other bifidobacteria [47].

There are limitations to this study that should be noted. The main one is the small research sample, including 54 subjects. The group of men in particular was small. This was due to the fact that the study protocol assumed the inclusion of subjects of similar height, age, and BMI in each subgroup (GP, GRD) to show small but potentially significant differences between them. More research is needed in this area on a larger group of people. The study was relatively short (12 weeks). We also have no guarantee that the participants of the intervention actually took yogurt in the recommended amounts and frequency. In order to verify the results achieved, long-term interventions with probiotic yogurts should be carried out in the future. Finally, the yogurts used were not made under controlled conditions by the authors of this study. This may have contributed to differences in the amount of live probiotic bacteria in a portion of the product, resulting from the conditions of transport and distribution of products in stores. However, we wanted to check the impact of commercially available probiotic yogurts used in everyday dietary practice.

## 5. Conclusions

Lifestyle modifications remain the primary form of therapy for obesity and associated metabolic disorders. One potentially effective strategy for treating obesity may be manipulation within the composition of the intestinal microflora. First of all, this therapy is safe due to the lack of reported side effects and the fact that it is well-tolerated and suitable for long-term use. Given the data mentioned above, the concept of manipulating the intestinal microbiota to increase the effectiveness of dietary treatment of obesity is currently gaining interest from researchers from around the world. The study showed that 12-week probiotic supplementation does not significantly improve the effectiveness of a reduction diet in the context of changing selected anthropometric parameters in obese people. However, given the main limits of this study (e.g., the statistical power, excluded subjects from analysis, and absence of assessment of observance), the above conclusion cannot be drawn from this study with sufficient confidence. Therefore, further long-term studies are needed to investigate the effect of probiotic supplementation on various anthropometric parameters in people with excessive body weight.

**Author Contributions:** Conceptualization, K.B., P.G.; methodology, K.B. and P.G.; formal analysis, K.B., P.G. and P.J.; investigation, K.B.; writing—original draft preparation, K.B.; writing—review and editing, K.B., P.G. and P.J.; visualization, K.B. and P.G.; supervision, P.G.; All authors have read and agreed to the published version of the manuscript.

**Funding:** This research received no external funding.

**Conflicts of Interest:** The authors declare no conflict of interest.

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
