# Peer review of "The Effect of Probiotic Yogurt Containing Lactobacillus Acidophilus LA-5 and Bifidobacterium Lactis BB-12 on Selected Anthropometric Parameters in Obese Individuals on an Energy-Restricted Diet: A Randomized, Controlled Trial"

_applsci, doi:10.3390/app10175830_

Round 1
Reviewer 1 Report
The authors of the manuscript took into account all comments. In this version the manuscript can be accepted in my opinion.
Author Response
Thank you very much
Reviewer 2 Report
The authors have replied the questions and revised this manuscript. It is now proper for publication in “Applied Sciences”.
Author Response
Thank you very much
Reviewer 3 Report
In the manuscript presented for review, the impact of consuming yogurt enriched with additional cultures of probiotic strains (La-5 and BB-12) on anthropometric parameters in obese people on a energy-restricted diet was assessed.
I suggest adding the names of the strains in the title → The Effect of Probiotic Yogurt Containing Lactobacillus acidophilus LA-5 and Bifidobacterium lactis BB-12 on Selected Anthropometric Parameters in Obese Individuals on an Energy-Restricted Diet: a Randomized, Controlled Trial.
I have no comments regarding the construction of the experiment. Of course, it has some limitations, but the authors themselves are aware of this (see paragraph in lines 288-299)."Introduction" introduces the subject well and comprehensively.
Chapter "Materials and methods" does not require changes - minor corrections in lines: 124 and 130.
In the following chapters, in my opinion, only the way of citing the sources of literature requires change (lines 205, 231, 247, 256, 268, 273).
In the "Discussion" section, the paragraph (in lines 239-252) on the effect of introducing probiotics into the diet on body weight and composition requires some additions. In the case of probiotics, in addition to the species name, the strain symbol is always given. All the above-mentioned comments were also made in the review mode to the pdf file.

Author Response
Dear Reviewer,
We thank for all remarks which enabled us to improve the quality of our manuscript. Our point-by-point responses to each of the comments are below.
All changes in the manuscript were highlighted. We hope that the revision in the text and our accompanying responses will be sufficient to make our manuscript suitable for publication in Applied Sciences. However, we are open for further remarks.
Sincerely yours,
Authors
Reviewer 3
Point 3.1: „I suggest adding the names of the strains in the title → The Effect of Probiotic Yogurt Containing Lactobacillus acidophilus LA-5 and Bifidobacterium lactis BB-12 on Selected Anthropometric Parameters in Obese Individuals on an Energy-Restricted Diet: a Randomized, Controlled Trial.”
Reply 3.1: We agree with the reviewer on this point. We have added the names of the strains in the title.
Point 3.2: „Chapter "Materials and methods" does not require changes - minor corrections in lines: 124 and 130.”
Reply 3.2: We have made minor corrections in lines: 124 and 130.”
Point 3.3: „In the following chapters, in my opinion, only the way of citing the sources of literature requires change (lines 205, 231, 247, 256, 268, 273).”
Reply 3.3: This is a valuable remark. We have change the way of citing the sources of literature.
Point 3.4: „In the "Discussion" section, the paragraph (in lines 239-252) on the effect of introducing probiotics into the diet on body weight and composition requires some additions. In the case of probiotics, in addition to the species name, the strain symbol is always given. All the above-mentioned comments were also made in the review mode to the pdf file.”
Reply 3.4: Thank you for this remark. We have added the strains symbols in lines 239-252.
This manuscript is a resubmission of an earlier submission. The following is a list of the peer review reports and author responses from that submission.
Round 1
Reviewer 1 Report
As stated by the authors the main limits of this study are the statistical power, per protocol analysis (4 subjects excuded from analysis) and absence of assessment of observance. Consequently, the conlusion that the product is not effective cannot be drawn from this study with sufficient confidence.
Authors should have expressed how they conceived the statistical hypothesis of the trial (which is obviously a pilot one).
The discussion is very general and many probiotic genera and strains are quoted which suggest to readers that generalisation of results from one product to another might be possible.
Reviewer 2 Report
The manuscript presents an interesting elaboration of the topic, although the authors stated that "Consumption of yogurt containing LA-5 and BB-12 does not significantly improve anthropometric parameters in obese patients". Please explain a few doubtful aspects.
Line 40-50: Please specify whether the proportion of Firmicutes to Bacteroidetes in obese people is positive or negative.
Line 118: Why did the authors decide on this particular yogurt? Does it differ significantly in composition from other commonly available yogurt of this type? Why did the authors decide to include yogurt in their diet and not probiotics in the form of capsules?
Results: What could have caused the lack of the assumed effect of yogurt supplementation?
Reviewer 3 Report
- Is there any scientific evidence showing the efficacy of yogurt containing LA-5 and BB-12 on treatment of obesity?
- In Section 2.3 (line 122), does the yogurt used in the present study contain “added sugar”?
- The authors used “hypocaloric diet” in group GP and “balanced energy-restricted diet” in group GRD. Is there any difference between these two diets?
- In addition to diet, “physical activity” and “fecal microbiota” are also important parameters in this study. Did the authors evaluate these two parameters?
- The authors should discuss in more detail why yogurt containing LA-5 and BB-12 failed to improve anthropometric parameters in obese patients.
- Figure 1 is not clear.
